# Senescence of IPF Lung Fibroblasts Disrupt Alveolar Epithelial Cell Proliferation and Promote Migration in Wound Healing

**DOI:** 10.3390/pharmaceutics12040389

**Published:** 2020-04-24

**Authors:** Kaj E. C. Blokland, David W. Waters, Michael Schuliga, Jane Read, Simon D. Pouwels, Christopher L. Grainge, Jade Jaffar, Glen Westall, Steven E. Mutsaers, Cecilia M. Prêle, Janette K. Burgess, Darryl A. Knight

**Affiliations:** 1School of Biomedical Sciences and Pharmacy, University of Newcastle, Callaghan, NSW 2308, AustraliaMichael.Schuliga@newcastle.edu.au (M.S.); jane.read@newcastle.edu.au (J.R.); darryl.knight@newcastle.edu.au (D.A.K.); 2National Health and Medical Research Council Centre of Research Excellence in Pulmonary Fibrosis, Camperdown, NSW 2050, Australia; christopher.grainge@newcastle.edu.au; 3University of Groningen, University Medical Center Groningen, Department of Pathology and Medical Biology, 9713 GZ Groningen, The Netherlands; j.k.burgess@umcg.nl; 4University of Groningen, University Medical Center Groningen, Groningen Research Institute for Asthma and COPD, 9713 GZ Groningen, The Netherlands; s.d.pouwels@umcg.nl; 5University of Groningen, University Medical Center Groningen, Department of Lung Diseases, 9713 GZ Groningen, The Netherlands; 6School of Medicine and Public Health, University of Newcastle, Callaghan, NSW 2308, Australia; 7Allergy, Immunology and Respiratory Medicine, Alfred Hospital, Prahran, Vic 3004, Australia; jade.jaffar@monash.edu (J.J.); g.westall@alfred.org.au (G.W.); 8Centre for Cell Therapy and Regenerative Medicine, School of Biomedical Sciences University of Western Australia, Crawley, WA 6009, Australia; steven.mutsaers@uwa.edu.au (S.E.M.); cecilia.prele@uwa.edu.au (C.M.P.); 9Institute for Respiratory Health, University of Western Australia, Nedlands, WA 6009, Australia; 10Providence Health Care Research Institute, Vancouver, BC V6Z 1Y6, Canada

**Keywords:** senescence, fibroblasts, alveolar epithelial cell, fibrosis, aberrant repair, cell-cycle inhibition

## Abstract

Idiopathic pulmonary fibrosis (IPF) is a progressive lung disease marked by excessive accumulation of lung fibroblasts (LFs) and collagen in the lung parenchyma. The mechanisms that underlie IPF pathophysiology are thought to reflect repeated alveolar epithelial injury leading to an aberrant wound repair response. Recent work has shown that IPF-LFs display increased characteristics of senescence including growth arrest and a senescence-associated secretory phenotype (SASP) suggesting that senescent LFs contribute to dysfunctional wound repair process. Here, we investigated the influence of senescent LFs on alveolar epithelial cell repair responses in a co-culture system. Alveolar epithelial cell proliferation was attenuated when in co-culture with cells or conditioned media from, senescence-induced control LFs or IPF-LFs. Cell-cycle analyses showed that a larger number of epithelial cells were arrested in G2/M phase when co-cultured with IPF-LFs, than in monoculture. Paradoxically, the presence of LFs resulted in increased A549 migration after mechanical injury. Our data suggest that senescent LFs may contribute to aberrant re-epithelialization by inhibiting proliferation in IPF.

## 1. Introduction

Idiopathic pulmonary fibrosis (IPF) is a devastating fibrosing interstitial lung disease (ILD) of unknown etiology with a survival of 3 to 5 years after diagnosis. IPF is associated with increasing age, with two-third of patients being over 60 years old at the time of diagnosis [1,2]. The mechanisms that underlie IPF pathophysiology remain unclear but recent theories have moved away from the long standing paradigm of chronic inflammation to one that supports recurrent alveolar epithelial cell injury followed by aberrant wound repair [3,4]. The injured alveolar epithelium secretes growth factors, cytokines and chemokines that autocrine promotes epithelial cell migration, proliferation and differentiation of type II cells into type I alveolar epithelial cells. In IPF, the ability of type II alveolar epithelial cells to migrate, proliferate and differentiate is compromised [5]. Dysfunctional repair could be attributed to excessive loss of alveolar epithelial cell loss by apoptosis which seems to be a feature of IFP. Secretion of factors from damaged alveolar epithelium also enables paracrine control of resident fibroblasts such as proliferation, chemotaxis and deposition of extracellular matrix [6]. The wound healing cascade that follows, orchestrated by lung fibroblasts (LFs), does not follow the normal repair process of healthy tissue; instead of achieving wound resolution, a mass of LFs accumulates at the site of injury forming fibroblastic foci, which are a major pathological feature of the disease [7,8]. The LFs in fibroblastic foci secrete exaggerated amounts of extracellular matrix proteins, which results in accumulation in the alveolar and interstitial space leading to destruction of lung parenchyma [9].

IPF presents mainly in the elderly, implying that aging and cellular senescence contributes to disease progression [10,11,12]. Senescent cells have been increasingly associated with structural changes seen in aging and several studies report that IPF is characterized by increased senescence in both LFs and alveolar epithelial cells [13,14,15,16]. Senescence occurs as a response to excessive stress locking the cells into cell-cycle arrest. Several stress factors can induce premature senescence including telomere attrition, oxidative stress or mitochondrial dysfunction [15,17,18]. Cells that undergo (premature) senescence are characterized by irreversible cell-cycle arrest and induction of anti-apoptotic genes, but more importantly, senescent cells remain metabolically active leading to the development of a dynamic pro-inflammatory secretory profile known as the “senescence-associated secretory phenotype” (SASP) [19]. The composition of the secretory profile is dynamic and dependent on the senescence inducer and cell origin [20]. Nevertheless, most SASP profiles will contain several pro-inflammatory and pro-fibrotic cytokines including CXCL1, GM-CSF, M-CSF, CXCL-8, IL-6, and TGF-β [21,22]. Through the SASP senescent cells are able to impact neighboring cells and the local microenvironment [13,14]. Here we aimed to investigate the role of senescent LFs on A549 alveolar epithelial cell proliferation and migration during wound healing using a co-culture model.

## 2. Materials and Methods

### 2.1. Human Lung Tissue

Primary LFs were obtained from lung tissue resections donated by patients who provided informed written consent from the John Hunter Hospital in accordance with HNEHREC 16/07/20/5.03. Additional primary LFs were obtained from the Alfred Lung Fibrosis Biobank (Alfred Hospital, Melbourne, Australia) under ethical approval #336/13 following NHMRC guidelines (26 November 2013). Primary LFs in culture were established as described before [23].

### 2.2. Co-Culture of LFs with Alveolar Epithelial Cells

Fibroblasts from non-ILD controls (Ctrl-LFs) or IPF patients (IPF-LFs) were co-cultured with A549 alveolar epithelial cells (American Type Culture Collection) separated by Corning Transwell Cell Culture Inserts with 0.4 µM pore size in a 12-well plate. Both LFs and A549 cells were maintained in Dulbecco’s Modified Eagle’s Medium (DMEM) GlutaMAX Low Glucose (Thermo Scientific, Scoresby, VIC, Australia) supplemented with 25 mM HEPES, 10% Fetal Bovine Serum (FBS), penicillin (100 U/mL) and streptomycin (100 µg/mL) (All from Sigma-Aldrich, Castle Hill, NSW, Australia). Primary LFs were used between passage 2 and 6 while A549 cells were used between passage 21 and 30. Fifty thousand LFs were seeded in a 12-well plate and allowed to adhere and recover for 48 h before being serum starved (DMEM with 0.4% FBS) for 48 h. To induce senescence, LFs were exposed to 150 µM hydrogen peroxide (H_2_O_2_) for 2 h followed by 3 days of incubation in DMEM with 0.4% FBS [24]. On the same day as LF senescent induction, 30,000 A549 cells were seeded on a transwell insert in DMEM with 10% FBS and allowed to recover for 48 h. The media was then replenished with DMEM containing 0.4% FBS, in which the cells were maintained for an additional 72 h before co-culture or receiving LF conditioned media (CM). To co-culture, transwell inserts containing A549 cells were transferred on top of the LFs and fresh DMEM with 5% FBS was added to stimulate A549 cell proliferation. After 48 h of co-culture or being maintained in LFs CM, A549 cells were harvested for cell enumeration and cell-cycle analysis.

For the migration assay a confluent layer of A549 cells was exposed to 0.5 µM mitomycin C (MMC) (Sigma-Aldrich) for 2 h followed by generation of a scratch-wound using a p20 pipette tip, before washing twice with PBS. Transwell inserts containing wounded A549 cells were transferred on top of the LFs stimulated as described before and fresh DMEM with 5% FBS was added to the apical side of the membrane. Images were taken using an inverted microscope (Leica DMIL LED, Amsterdam, The Netherlands) equipped with a Leica MC120 HD camera. ImageJ was used to measure the wound area to calculate a rate of wound closure [25].

### 2.3. Cell Enumeration

Transwell inserts containing A549 cells were transferred to a new 12-well plate and harvested using a mixture of 0.5% trypsin and 2 mg/mL collagenase Type I in Hank’s Balanced Salt Solution containing calcium and magnesium but without phenol red (HBSS) (All from Sigma-Aldrich) for 20 min at 37 °C, 5% CO_2_. The trypsin/collagenase solution was inactivated using PBS containing 5 mM ethylenediaminetetraacetic acid (EDTA) and 2.5% *v*/*v* FBS. An A549 cell aliquot was transferred to a new tube, stained using trypan blue (Sigma-Aldrich) and manually counted using a hemocytometer.

### 2.4. Immunoblotting

Cell lysates were measured using the BCA assay kit according to manufacturer specifications (Thermo Scientific) before 10 µg protein was subjected to SDS polyacrylamide gel electrophoresis followed by semi-dry transfer as described before [23]. Primary antibodies used were p21 (1:1000) (CST, #2946) Phospho-Rb (1:1500) (CST, #3590) and β-Actin (1:5000) (Abcam, #ab8227).

### 2.5. Cell-Cycle Analysis

Cell-cycle kinetics of A549 cells were evaluated using propidium iodide (PI) (Sigma-Aldrich) detection by fluorescent-activated cell sorting analysis. Cells were harvested after co-culture and fixed in ice-cold 70% ethanol for 1 h. After washing with HBSS, 50 µL ribonuclease I (100 µg/mL) was added and incubated for 30 min at room temperature. PI (50 µg/mL) was added to the dissociated cells before being incubated for 10 min on ice. Twenty thousand events were collected and analyzed on a FACSCanto II (Becton Dickinson, Macquarie Park, Australia). Cell-cycle kinetics was quantified using FlowJo™ software (Version 10, FlowJo LLC, Ashland, OR, USA).

### 2.6. Statistical Analysis

Statistical analyses were performed using GraphPad Prism (Version 8, GraphPad Software, La Jolla, CA, USA) and data presented as mean ± SD with each point representing a different donor. Statistical analysis was evaluated using Wilcoxon matched-pair signed rank test for comparison between stimulated and unstimulated conditions. Unpaired nonparametric Mann–Whitney test was used to compare Ctrl-LFs with IPF-LFs. Data were considered statistically significant at *p* < 0.05.

## 3. Results

### 3.1. Senescent LFs Reduce the Proliferation of Alveolar Epithelial Cells in Co-Culture

We investigated the effect of Ctrl-LFs and IPF-LFs with or without H_2_O_2_ stimulation on A549 cell proliferation in co-culture (Figure 1). Table 1 characterized the fibroblast donors used for this study. Samples were chosen at random for any assay. Co-culture with Ctrl-LFs did not reduce A549 cell proliferation compared to A549 monoculture. However, co-culture with H_2_O_2_-exposed (senescent) Ctrl-LFs significantly reduced A549 proliferation (78.7 ± 12.1%) when compared to untreated Ctrl-LFs (*p* = 0.0313). IPF-LFs at baseline decreased A549 cell proliferation (87.1 ± 8.5%) when compared to Ctrl-LF co-culture (*p* = 0.0173) and A549 monoculture. Interestingly, H_2_O_2_ stimulated IPF-LFs further exaggerated this effect and strongly reduced proliferation (62.2 ± 8.1%) compared to all other mono- or co-cultures (*p* < 0.05). These data indicate that A549 cell proliferation is inhibited by senescent-induced Ctrl-LFs or IPF-LFs in co-culture.

### 3.2. Conditioned Medium from Senescent-Induced LFs Reduces the Proliferation of Alveolar Epithelial Cells

Next, we evaluated whether the anti-proliferative effect of senescent induced Ctrl-LFs and IPF-LFs on A549 cells are caused by factor(s) actively secreted as part of the SASP. A549 cells were incubated with conditioned medium (CM) from Ctrl-LFs and IPF-LFs with or without H_2_O_2_ stimulation and proliferation measured (Figure 2). CM from untreated Ctrl-LFs had no significant effect on A549 proliferation compared to control. However, CM from H_2_O_2_-stimulated Ctrl-LFs significantly reduce A549 proliferation (85.7 ± 24.6%) but to a lesser extent than in co-culture system (*p* = 0.0313). Similarly, only A549 cells incubated with CM from H_2_O_2_ treated IPF-LFs demonstrated reduced proliferation (82.4 ± 26%) (*p* = 0.0313). These results suggest that inhibition of A549 cell proliferation by senescent LFs is dependent, at least partly on secreted factors as part of the SASP.

### 3.3. Co-Culture with IPF-LFs Induces Alveolar Epithelial Cell-Cycle Arrest

Cells that undergo senescence show signs of permanent cell-cycle arrest. Cell-cycle distribution was assessed using PI staining of A549 cells at baseline or in co-culture with Ctrl-LFs and IPF-LFs with or without H_2_O_2_ stimulation. (Figure 3). Here, it was shown that co-cultures with IPF-LFs and A549 cells significantly increased the number of alveolar epithelial cells in G2/M phase compared to A549 cells at baseline (*p* = 0.0455) or Ctrl-LFs (*p* = 0.0303). Co-cultures with H_2_O_2_ treated Ctrl-LFs and IPF-LFs with A549 cells shown similar impact on G2/M phase as IPF-LFs alone. Co-culture with Ctrl-LFs had no impact on distribution of cell-cycle phase compared to A549 monoculture.

### 3.4. Primary Lung Fibroblasts Increase Alveolar Epithelial Cell Migration in Co-Culture

Next, the effect of senescent LFs on A549 cell wound closure after mechanical injury was studied (Figure 4). To assess migration only during wound closure we treated A549 cells prior wounding with MMC to inhibit proliferation after mechanical injury. Optimal MMC concentration was chosen based on cell-cycle analysis and p21/pRb protein expression (Figure 4C,D). A549 cells alone did not completely close the wound within 48 h (Figure 4A). No combination of cells and treatments closed the wound by 24 h but the wound was closed by 48 h when A549 cells were co-cultured with Ctrl-LFs or IPF-LFs irrespective of whether or not they were treated with H_2_O_2_ (Figure 4).

## 4. Discussion

IPF is characterized by accumulation of extracellular matrix as a result of an impaired wound repair response and an increase in senescence. There is accumulating evidence that fibroblast senescence contributes to IPF pathology, but its role in dysfunctional alveolar epithelial repair remains to be elucidated. By using a co-culture model with primary LFs, we were able to assess the effect of Ctrl-LFs, IPF-LFs and oxidative-stress induced senescence on the proliferation and migration of A549 cells in co-culture. Hydrogen peroxide has been described and used for nearly three decades to induce oxidative-stress induced senescence in primary human fibroblasts with minimal cytotoxicity [26,27,28]. Recently a publication by Waters et al., 2019 from our group characterized in detail fibroblast stress-induced senescence after H_2_O_2_ treatment [23]. A single treatment with 150 µM H_2_O_2_ for 2 h resulted in increased markers of senescence such as p21 expression, nuclear phospho-p53, increased IL-6 secretion and cytoplasmic SA-β-Gal in primary human lung fibroblasts. Mitochondrial characterizations such as basal respiration, proton leak and superoxide generation were also significantly increased after H_2_O_2_ exposure. All of which are characteristics of senescence [17]. We showed that A549 cell numbers in co-culture were strongly attenuated by senescent Ctrl-LFs. We further found that IPF-LFs also strongly reduce A549 cell proliferation; previous studies including our own have shown that IPF-LFs are more senescent-like than control LFs [15,29]. Our current findings suggest that senescent LFs may orchestrate an impaired repair response by limiting proliferation of neighboring alveolar epithelial cells in IPF. Whether this reduction in proliferative potential is irreversible remains unexplored. Interestingly, stimulation of IPF-LFs with H_2_O_2_ enhanced the inhibition of A549 cell proliferation. During in vitro expansion of IPF-LFs there is active selection for proliferating cells. This could explain why some IPF-LF cultures react to treatment with H_2_O_2_ leading to a higher number of senescent cells and more pronounced inhibition in co-culture. These data suggest that the mechanisms by which IPF-LFs reduce A549 cell proliferation is independent to that of oxidative-stress induced senescent IPF-LFs. In vitro several ways to induce senescence are known; however, it is unknown whether these different types of senescence also occur in vivo [17]. How senescence of LFs can influence aberrant re-epithelialization and especially the antiproliferative effect of alveolar epithelial cells in IPF is not clear. On the one hand several studies describe the findings that proliferation and hyperplasia of alveolar epithelial cells contributes to the profibrotic environment [30]. While others demonstrated that epithelial cell senescence and apoptosis is a potent inducer of IPF [31,32,33].

It has been shown that the SASP is a core characteristic of IPF-LFs, comprising of pro-inflammatory and pro-fibrotic factors that influence the local micro-environment. To explore possible paracrine influences of the SASP, we exposed A549 cells to CM from senescent-induced LFs or IPF-LFs. We found that CM from senescent induced LFs (either control or IPF) reduced A549 cell proliferation. These findings support the hypothesis that senescent LFs actively secrete factors as part of the SASP affecting the local micro-environment. Interestingly, the difference in the anti-proliferative effects between the IPF-LF co-culture and CM experiments, suggest the factor released from IPF-LFs responsible for reducing A549 cell proliferation is a factor that is rapidly produced and then inactivated. Using a CM model, we eliminate potential contact-dependent crosstalk between the LFs and A549 cells. The senescent LFs may produce a factor that stimulates the A549 cells to produce a messenger molecule that in turn induces the LFs to produce an anti-proliferative compound creating a feedback system; and highlighting the importance of crosstalk [34].

Cells that undergo senescence and develop a SASP can mediate paracrine transmission of senescence to neighboring cells [35]. Using our model, we measured an increased number of A549 cells in G2/M cell-cycle arrest when in co-culture with IPF-LFs. A similar increase in G2/M cell-cycle arrest of A549 cells co-cultured with both Ctrl-LFs (*n* = 3) and IPF-LFs (*n* = 2) treated with H_2_O_2_ was also observed. This suggests that senescent Ctrl-LFs and IPF-LFs are able to induce senescence in neighboring epithelial cells as demonstrated by the reduced proliferation of A549 cells in our co-culture experiments. However, how this transmission occurs is not clear. It might be SASP and/or through the release of reactive oxygen species as mitochondrial dysfunction is a characteristic of the senescent phenotype [36]. Nelson and colleagues demonstrated that senescent fibroblasts can induce DNA damage in bystander cells by the release of reactive oxygen species [37]. Whether the increase in G2/M cell-cycle arrest we measured is irreversible remains unclear as we did not examine markers of senescence such as senescence-associated β-galactosidase, p21 gene expression or DNA damage response (DDR) in A549 cells.

During re-epithelization, type II alveolar epithelial cells proliferate, differentiate into type I cells and migrate over the remodeled matrix to quickly restore normal homeostasis. In IPF this process is impaired and it a number of profibrogenic mediators seems to be implicated in this dysfunctional response characterized by excessive alveolar epithelial cell loss [5]. Having demonstrated that senescent LFs were able to reduce proliferation, we exposed A549 cells to mechanical injury to assess their migratory capacity in the presence of senescent-induced Ctrl-LFs and IPF-LFs or A549 cells alone. During wound closure we observed that the presence of LFs, regardless of disease status or senescent-induced phenotype, created a positive migration environment. The lack of difference between migration induced by Ctrl-LFs and IPF-LFs is compelling as several studies have shown the importance of senescent cells in wound healing [38]. Krizhanovsky and colleagues illustrated in an animal model of liver injury that accumulation of senescent cells prevented fibrotic tissue formation and promoted resolution of fibrosis [39]. Moreover, our observation that IPF-LFs have a positive influence on epithelial migration is supported by a study of Prasad and colleagues, in which it was shown that IPF-LFs promote alveolar epithelial cell migration while Ctrl-LFs migrate into the damaged area during wound-healing [40]. Our results highlight the potential difference in mechanisms between reducing proliferation and promoting migration. Although great progress has been made in understanding senescence and its role in IPF, much remains unclear as to how it contributes to aberrant re-epithelialization in IPF.

While we recognize the translational limitations of using the adeno-carcinoma cell-line A549 as model to study epithelial-fibroblast crosstalk in relation to IPF, it remains the best and most robust alveolar epithelial cell-line to date to study basic biological processes. To identify potential mechanisms by which senescent cells impact on alveolar epithelial cells modulation of known SASP factors such as IL-6, IL-1β, PDGF, FGFb, PGE_2_ and ROS would be the focus of future studies [41,42]. Interestingly, STAT3 has been reported to be involved in fibroblast senescence but a recent publication has highlighted the potential involvement in epithelial cell senescence [43]. This is the first study to show that senescent LFs are able to reduce alveolar epithelial cell proliferation in a co-culture system. In line with previous observations, senescent LFs secrete components as part of the SASP that contribute to the spread of senescence to neighboring cells. The IPF-LFs are inherently different in their secretory profile which may have implications for understanding the diseased biology in patients’ tissues.

## 5. Conclusions

We have shown that senescent-induced and IPF-LFs are able to reduce alveolar epithelial cell proliferation in co-culture and that this is independent of crosstalk between LF’s and alveolar epithelial cells. Increase in cell-cycle arrest in alveolar epithelial cells suggest a role for the secretory profile in spreading senescence to neighboring cells. Furthermore, we have shown that migration of alveolar epithelial cells during wound repair is not impacted by disease status or induction of senescence. Overall, our study highlights the importance of senescent LFs and difference in mechanism of reducing proliferation and increase in migration during injury, but the role exact of re-epithelialization in IPF remains unclear.

## Figures and Tables

**Figure 1 pharmaceutics-12-00389-f001:**
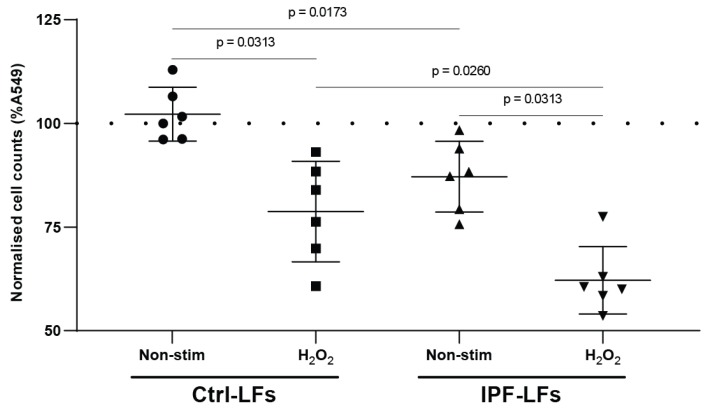
Senescent LFs reduce proliferation of A549 cells in co-culture. A549 cells were co-cultured in the presence of Ctrl-LFs (*n* = 6) or IPF-LFs (*n* = 6). Fibroblast senescence was induced by stimulation with 150 μM H_2_O_2_ for 2 h followed by incubation for 72 h in low-serum DMEM, and afterwards co-cultured for 48 h. Proliferative potential of A549 cells was measured by cell enumeration. All data were normalized to A549 cell baseline growth (dotted line, 100%) and expressed as mean ± SD, *p* < 0.05 was considered statistically significant, Wilcoxon matched-pairs rank test for non-stimulated and H_2_O_2_; Mann–Whitney U for Ctrl-LFs vs. IPF-LFs at baseline.

**Figure 2 pharmaceutics-12-00389-f002:**
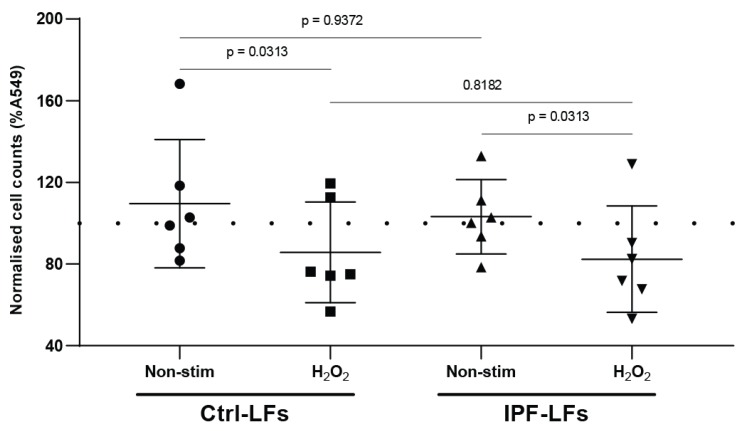
Conditioned medium from senescent LFs reduced proliferation of A549 cells. A549 cells were cultured in conditioned media from Ctrl-LFs (*n* = 6) or IPF-LFs (*n* = 6) fibroblasts. Fibroblast senescence was induced by stimulation with 150 μM H_2_O_2_ for 2 hrs followed by recovery for 72 h, conditioned media transfer and cultured for 48 h. Proliferative potential of alveolar epithelial cells was measured by cell enumeration. All data were normalized to A549 baseline growth (dotted line, 100%) and expressed as mean ± SD, *p* < 0.05 was considered statistically significant, Wilcoxon matched-pairs rank test for non-stimulated and H_2_O_2_; Mann–Whitney U for Ctrl-LFs vs. IPF-LFs at baseline.

**Figure 3 pharmaceutics-12-00389-f003:**
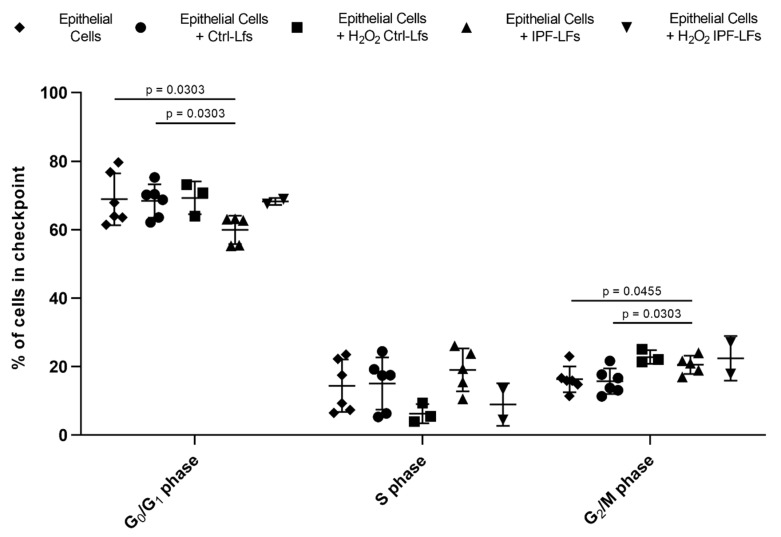
Co-culture of A549 cells with IPF-LFs increased cell-cycle arrest in G2/M phase. A549 cells were co-cultured with Ctrl-LFs(*n* = 6) and H_2_O_2_ treated Ctrl-LFs (*n* = 3) or IPF-LFs(*n* = 6) and H_2_O_2_ treated IPF-LFs (*n* = 2) for up to 48 h after cells were fixed and stained for DNA content using PI followed by FACS analysis. Fibroblast senescence was induced by stimulation with 150 μM H_2_O_2_ for 2 h followed by incubation for 72 h in low-serum DMEM, and afterwards co-cultured for 48 h. Diamonds represents A549 cell baseline cell-cycle profile, circles represents co-culture with Ctrl-LFs while squares represents co-culture with H_2_O_2_ treated Ctrl-LFs. Up-pointing triangle represents co-culture with IPF-LFs and down-pointed triangle represents co-culture with H_2_O_2_ treated IPF-LFs. Cell-cycle data were expressed as relative cell-cycle phase (%) ± SD.

**Figure 4 pharmaceutics-12-00389-f004:**
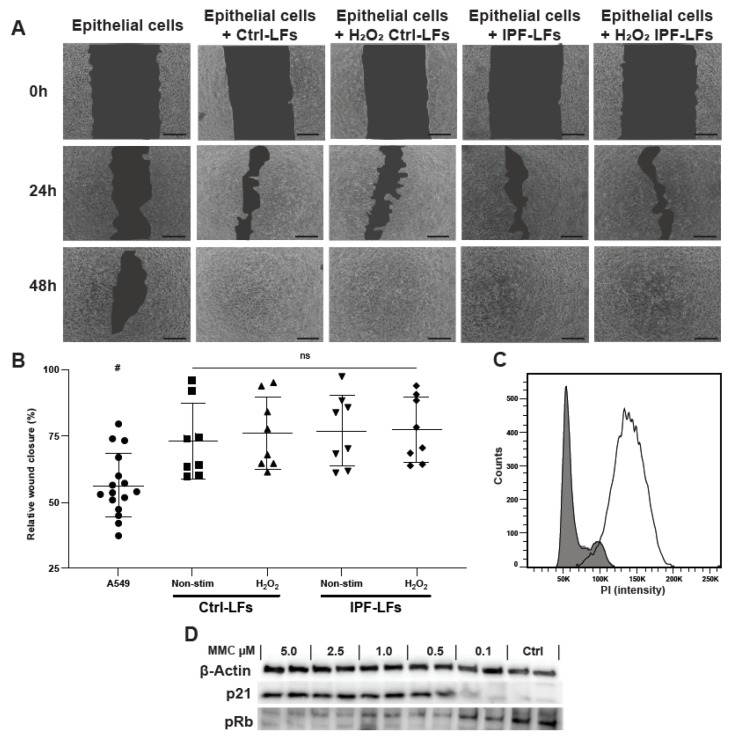
Fibroblasts increase alveolar epithelial cell migration in co-culture. (**A**) A549 cells were grown to confluence, treated with 0.5 μM MMC for 2 h before a scratch-wound was introduced. Fibroblast senescence was induced by stimulation with 150 μM H_2_O_2_ for 2 h. (**A**) The mechanically injured A549 cells were cultured alone (*n* = 16) or in combination with Ctrl-LFs (*n* = 8) or IPF-LFs (*n* = 8), and migration was followed up for 48h. (**B**) The relative rate of wound closure after 24 h was calculated based on surface area of the scratch using ImageJ. Data are expressed as relative wound closure in percentage after 24 h ± SD. # A549 cell wound closure at 24 h was tested significant different compared to all other conditions using Mann–Whitney U test for each co-culture condition (*p* < 0.05). (**C**,**D**) MMC concentration was titrated followed by cell-cycle and immunoblotting to select for optimal concentration. Scale bar: 250 nm.

**Table 1 pharmaceutics-12-00389-t001:** Characteristics of fibroblast donors used in this study. N/A = data not available. Mean age of non-ILD donors 54 years and IPF donors 59 years of age. Fibroblast samples were chosen at random for any assay.

Donor #	Sex	Age	Diagnosis	Smoking History	Pack Years
1	N/A	39	Donor	Ex	15
2	M	69	Donor	Ex	20
3	N/A	35	Donor	N/A	N/A
4	F	67	Donor	N/A	N/A
5	F	61	Donor	N/A	N/A
6	F	66	Donor	Ex	28
7	M	65	IPF	Ex	40
8	M	63	IPF	Ex	20
9	F	56	IPF	Never	0
10	M	57	IPF	Ex	48
11	F	59	IPF	Ex	60
12	M	43	IPF	Ex	7
13	M	70	IPF	never	0
14	M	54	IPF	Ex	N/A

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
