# Peer review of "Senescence of IPF Lung Fibroblasts Disrupt Alveolar Epithelial Cell Proliferation and Promote Migration in Wound Healing"

_pharmaceutics, 2020, doi:10.3390/pharmaceutics12040389_

Round 1

Reviewer 1 Report

The manuacript titled ‘Senescence of IPF lung fibroblasts disrupt alveolar epithelial cell proliferation and promote migration in wound healing’ demonstrated that Alveolar epithelial cell proliferation slowed down by senescence-induced control LFs or IPF-LFs and the presence of LFs resulted in increased A549 migration after mechanical injury.

The manuscript has a few gaps that need to be addressed before considering for publication.

  1. The proloferation augmentatation needs further testing and conclude what is the possible mechanism of action (inhibition of any molecular responses). Althouh the authors attempted to explain the mechanism with support from literature, concrete data generation is needed.
  2. It was mentioned that presence of LFs resulted in increased A549 migration after mechanical injury. It is true in all cases except untreated. What is the purpose of H2O2 induction in the experiment? The experiment design may need to be refined and explore more molecular responses quantified by qPCR.

Reviewer 2 Report

The manuscript entitle « senescence of IPF lung fibroblasts disrupt alveolar epithelial cell proliferation and promote migration in wound healing » written by Blokland et al is interesting and addresses an important question in the pathophysiology of IPF. The manuscript is well written and experiments in adequation with the question asked. However I have some questions about the experimental design.They demonstrate that co-culture of either control or IPF lung fibroblasts with alveolar epithelial cells decrease alveolar cell proliferation associated with a cell cycle arrest in G2/M phase and an increase in cell migration. They demonstrate that cell proliferation arrest as well as the cell cycle arrest in G2/M phase is more pronounced when alveolar epithelial cells were cultured with lung fibroblasts from IPF patients. However no difference is observed in wound closure. 

I have some questions about these results:

 What appends in alveolar epithelial cells? Could you show some molecular markers of either apoptosis (caspase 3)? Senescence (p21, p16, b-gal staining?)? Proliferation (Ki67, PCNA)? Migration (a-sma)? 

How did you evaluate the effect of H2O2 on the senescence of lung fibroblasts? Is H2O2 cytotoxic? What is the mortality rate? This question is essentialIs there a difference between fibroblasts from non IPF and IPF patients when treated with H2O2?

Did they secrete more SASP that would explain the difference?  Why don’t you show the impact of IPF on alveolar cell cycle when treated with H2O2? Why don’t you show a difference in cell migration between IPF and non IPF condition?

Another experiment with Boyden chamber should strengthen your result. Did you have an idea for the soluble factor which could be implicated in this effect?

Could you propose some new experiments to show the difference and or the similitudes between IPF and non IPF fibroblasts and/or treated or not with H2O2?

Round 2

Reviewer 1 Report

The responses from authors are backed by literature. However, without studying proper mechanism of action and ruling out the effects of components on cell migration, the manuscript is not comprehensive. 

In addition, the rationale and experimental design for 'Primary lung fibroblasts effects on alveolar epithelial cell migration in co-culture' is clear in the manuscript. 

Reviewer 2 Report

After reading the authors response to the reviewer comments, some concerns remains unclear.

I believe that senescence markers in AEC are needed to convince the reader that alveolar cells are senescent. This point should be better argued.

The response of the authors to the second point of the reviewer should be added to the manuscript.

Concerning the third point, the authors should proposed to study modulation of one or more molecular markers to have an idea of the impact of either the SASP or senescent fibroblast alone, on the alveolar cell response.

Concerning the scratch wound assay and the interest of the Boyden chamber, I’m completely convinced that the use of Boyden chamber or a wound assay without scratch will not give the same answer. Indeed, a scratch induce the release of numerous factors from injured cells which are different from senescent cells

Finally concerning the last point. Remember that A549 are alveolar cell adenocarcinomas and not bronchial epithelial cells. Concerning the repair after injury, the function of each cell type is completely different and it is inconceivable to transpose your results on this.

Round 3

Reviewer 1 Report

Authors attempted to address the comments and improve the manuscript. Without mechanism of action and concrete data to support it, the results are not supporting the claims in the title. 

Further studies needed before publishing the manuscript. 

Reviewer 2 Report

Some concerns remains undemonstrated but the authors have improved the discussion.

the authors' arguments regarding the missing manipulations are acceptable and included in the discussion.

No further commens